

# Ensemble machine learning reveals key features for diabetes duration from electronic health records

Gabriel Cerono[1] and Davide Chicco[2,3]

[1] Department of Neurology, University of California San Francisco, San Francisco, CA, USA
[2] Institute of Health Policy Management and Evaluation, University of Toronto, Toronto, Canada
[3] Dipartimento di Informatica Sistemistica e Comunicazione, Università di Milano-Bicocca, Milan, Italy

## ABSTRACT

Diabetes is a metabolic disorder that affects more than 420 million of people worldwide, and it is caused by the presence of a high level of sugar in blood for a long period. Diabetes can have serious long-term health consequences, such as cardiovascular diseases, strokes, chronic kidney diseases, foot ulcers, retinopathy, and others. Even if common, this disease is uneasy to spot, because it often comes with no symptoms. Especially for diabetes type 2, that happens mainly in the adults, knowing how long the diabetes has been present for a patient can have a strong impact on the treatment they can receive. This information, although pivotal, might be absent: for some patients, in fact, the year when they received the diabetes diagnosis might be well-known, but the year of the disease unset might be unknown. In this context, machine learning applied to electronic health records can be an effective tool to predict the past duration of diabetes for a patient. In this study, we applied a regression analysis based on several computational intelligence methods to a dataset of electronic health records of 73 patients with diabetes type 1 with 20 variables and another dataset of records of 400 patients of diabetes type 2 with 49 variables. Among the algorithms applied, Random Forests was able to outperform the other ones and to efficiently predict diabetes duration for both the cohorts, with the regression performances measured through the coefficient of determination $R^2$. Afterwards, we applied the same method for feature ranking, and we detected the most relevant factors of the clinical records correlated with past diabetes duration: age, insulin intake, and body-mass index. Our study discoveries can have profound impact on clinical practice: when the information about the duration of diabetes of patient is missing, medical doctors can use our tool and focus on age, insulin intake, and body-mass index to infer this important aspect. Regarding limitations, unfortunately we were unable to find additional dataset of EHRs of patients with diabetes having the same variables of the two analyzed here, so we could not verify our findings on a validation cohort.

Corresponding author
Davide Chicco,
davide.chicco@gmail.com

## INTRODUCTION

Diabetes mellitus is group of metabolic diseases characterized by hyperglycemia and an epidemic affecting more than 420 million of people worldwide (*Chatterjee, Khunti & Davies, 2017*). Diabetes mellitus can be classified in two main types: type 1 (T1DM) and type 2 (T2DM). T2DM often occurs in older populations, accounting for 90% of total diabetes cases (*Sattar et al., 2019*), although it is increasingly seen in younger people (*Chen, Magliano & Zimmet, 2012*). T2DM appears with a gradual onset and is characterized by an impaired insulin metabolism due to dysfunctional beta pancreatic cells, or peripheral resistance to it, or both (*DeFronzo et al., 2015*). In contrast, T1DM has an acute clinical debut in childhood, and makes the patients suffer from lack of insulin production due to chronic autoimmune destruction of beta pancreatic cells. Latent autoimmune diabetes of adults (LADA) is a sub-variation of diabetes mellitus type 1 (*Djekic, Mouzeyan & Ipp, 2012*), that develops in people over 30 years old (*Naik, Brooks-Worrell & Palmer, 2009*), and differs from classical T1DM in its gradual clinical onset (*Isomaa et al., 1999*).

Diabetics patients are exposed to deleterious effects of hyperglycemia throughout the years, and their risk of suffering from multiple micro and macro-vascular complications increases overtime. Multiple randomized clinical trials have shown that an intensive control of glycemic levels greatly reduces the risk of experiencing these complications (*Control, of Diabetes Interventions & Group, 2005*). Adequate glycemic control becomes harder to achieve as the disease advances, and increasingly complex therapies accounting for multiple comorbidities are required in patients with long standing diabetes (*Longo et al., 2019*). Diabetic duration is therefore a critical risk factor when managing these patients. Unfortunately, this information is sometimes unknown as the disease can progress sub-clinically for years before a diagnosis is made.

Electronic health records (EHRs) have become an integral part of medical care (*Adane, Gizachew & Kendie, 2019*) providing doctors with reliable information that support clinical decisions. Analysis of the accumulated data of EHRs and the implementation of predictive models is pivotal for the advancement of medicine, as it could shed a light into hidden correlations that might not be evident or clear at first sight (*Štiglic et al., 2018*; *Benhamou, 2011*). Implementation of EHRs by medical teams have improved drug treatment intensification, monitoring and physiologic control in diabetic patients (*Reed et al., 2012*).

Regression analysis is a widely used statistical tool in health sciences, and it is employed to illustrate the relationship between explanatory variables and a target feature (*Liang & Zeger, 1993*). In this context, different clinical and laboratory variables can be of use to predict past diabetes duration. Classic linear regression is often limited by non-linearity relationships, heterogeneity of effects and high dimensionality; fortunately, machine learning regression techniques have been found to overcome these limitations (*Steele et al., 2018*; *Goldstein, Navar & Carter, 2016*).

The scientific literature shows that data mining models have demonstrated to be capable of managing different facets of diabetes mellitus, in the past. For example, *Bernardini et al. (2019)* identified patients with early insulin resistance from health record data

implementing a novel ensemble method and provided novel insights about the utilization of non-standard clinical risk factors to screen for early presentation of the disease. Machine learning techniques have predicted possible life-threatening hypoglycemic events during treatment (*Georga et al., 2013*), providing doctors with the capacity to tailor their treatment in this high risk population. Applied to data of EHRs of pregnant women, machine learning algorithms predicted the development of gestational diabetes, pointing out the need of a throughout screening regimen and early interventions in these patients (*Artzi et al., 2020*).

### Problem statement and motivation

Duration of diabetes is often unknown particularly for those patients who did not attend regular medical check-ups, and might have suffered from the disease for years before a diagnosis is made. In this group of patients, it is impossible to retrospectively know when the diabetes started. Recovering this information could be useful in foreseeing the evolution of the disease, the response to treatment, and the selection of proper screening methods (*Bax et al., 2007*; *Pham-Short et al., 2015*; *Thomas, Harvey & Owens, 2016*). In this context, supervised machine learning models can be used to discover past diabetes duration of the patients.

### Objective and novelty

The goal of our study is to predict the past duration of diabetes and then to detect the most predictive clinical variables. The novelty of our project lies in the usage of computational intelligence methods, together with recursive feature elimination and the coefficient of determination ($R^2$) metric.

### This study

Here, our approach was first to construct a regression model on data from two different sets of health records. The diabetes type 1 dataset (Takashi2019) contains 20 variables from 73 individuals, and the diabetes type 2 (AlOlaiwi2018) contains 49 variables, from 400 patients. Our work can be described in two parts. First, we developed various regression models to predict duration of diabetes using different machine learning algorithms, resulting in Random Forests (*Breiman, 2001*) being our top predictor. Second, we extended our analysis by generating a ranking of key features from both datasets utilizing our best predictor (Random Forest), to unveil correlations that may be concealed from classical statistical analysis. Our ranking concluded that age, body mass index, and insulin intake are key predictors of duration of diabetes on both populations. To the best of our knowledge, no study on the prediction of past diabetes duration exists in the scientific literature.

## DATASETS

For our analysis, we used two datasets, both made of electronic health records and publicly available online under the Creative Commons Attribution 4.0 International (CC BY 4.0) license: the Takashi2019 dataset of patients with diabetes type 1 (*Takashi et al., 2019*) and the AlOlaiwi2018 dataset of patients with diabetes type 2 (*AlOlaiwi, AlHarbi & Tourkmani, 2018*).

**Table 1  Meaning and measurement unit of the variables of the Takashi2019 diabetes type 1 dataset.**
Ug/ml: microgram per milliliter. kg/m$^2$ = kilogram per meter squared. pg/ml: picograms per milliliter.
ml/minutes/1.73 m$^2$: milliliters per minute per 1.73 m squared. m/s: meters per second. ng/ml: nanogram
per milliliter.

| Feature name | Measurement | Meaning |
|---|---|---|
| Added weight | kg | Calculated patient's weight |
| Adiponectin | Ug/ml | Serum adiponectin |
| Age | Years | Age of the patient at the medical check-up |
| Basal | Units of insulin | Daily basal dose of insulin. |
| BMI | kg/m$^2$ | Body mass index |
| Bodyfat | % | Bodyfat percentage |
| Bolus | Units of insulin | Daily bolus dose of insulin. |
| Duration of diabetes | Years | Duration of diabetes type 1 from onset until the medical check-up |
| eGFR | ml/minutes/1.73 m$^2$ | Estimated glomerular filtration rate |
| Free-test | pg/ml | Serum free testosterone concentration |
| Gait speed | m/s | Walking speed on a 5 m distance |
| Grip strength | kg | Grip strength measured using handheld dynamometers |
| HbA1c | % | Percentage of glycosylated hemoglobin |
| Insulin regimen | binary | MDI: multiple daily injections = 1; CSII: continous subcutaneus injections = 0 |
| Knee extension strength | kg | Knee extension strength measured using handheld dynamometers |
| OC | ng/ml | Total osteocalcin |
| Sex | Binary | male = 1; female = 0 |
| SMI | kg/m$^2$ | Skeletal muscle mass index |
| TDD | Units of insulin | Total daily dose of insulin |
| ucOC | ng/ml | Undercarboxilated osteocalcin |

## Diabetes type 1 dataset

The Takashi2019 dataset contains data of 73 diabetic patients. Each patient profile has 20
variables, including one that indicates the past duration of diabetes in years, that we use
as target variable (Table 1). The original data curators *Takashi et al. (2019)* collected these
data at the Osaka University Hospital and Osaka Police Hospital in July and August 2017,
and released them publicly in May 2019.

The Takashi2019 diabetes type 1 dataset features are related to clinical characteristics
of the patients (age, weight, body-mass index, sex, skeletal muscle mass index), or to
her/his well-being activity (gait speed, knee extension), or to blood test results (serum
adiponectin, testosterone concentration, hemoglobin, ostocalcin, underrcarboxilated
osteocalcin) (Table 1).

The patients of Takashi2019 diabetes type 1 dataset have an average weight of 63.35 kg
and an average age of 34.73 years (Table 2). Almost 70% of them are women and 30% are
men (Table 3).

**Table 2  Quantitative characteristics of the numeric features of the Takashi2019 diabetes type 1 dataset.**

| Numeric feature | Median | Mean | s.d. | Range |
|---|---|---|---|---|
| Added weight | 59.40 | 63.35 | 11.91 | [44.40, 104.90] |
| Adiponectin | 12.90 | 14.30 | 6.21 | [3.5, 32.3] |
| Age | 35.00 | 34.73 | 6.16 | [21, 48] |
| Basal | 14.84 | 16.23 | 8.08 | [0, 60.05] |
| BMI | 22.87 | 23.76 | 3.47 | [17.584, 35,54] |
| Body fat | 0.26 | 0.27 | 0.07 | [0.13, 0.48] |
| Bolus | 22.88 | 27.63 | 14.96 | [7.37, 93.94] |
| Duration of diabetes type 1 [target] | 26.00 | 25.68 | 7.33 | [10, 41] |
| eGFR | 92.74 | 92.86 | 14.06 | [50.7, 127.01] |
| Free-test | 1.30 | 4.24 | 5.0 | [0.4, 18.1] |
| Gait speed | 1.31 | 1.34 | 0.22 | [0.81, 2.00] |
| Grip strength | 30.20 | 32.08 | 8.77 | [16.79, 54.5] |
| HbA1c | 7.25 | 7.38 | 1.03 | [5.1, 10.7] |
| Knee extension strength | 20.00 | 20.59 | 5.85 | [8.70, 39.09] |
| OC | 14.80 | 16.25 | 7.89 | [6.4, 49.6] |
| SMI | 6.70 | 6.93 | 0.88 | [5.5, 9.2] |
| TDD | 40.00 | 43.87 | 19.85 | [15.7, 154.0] |
| ucOC | 3.25 | 4.17 | 3.26 | [0.53, 19.10] |

**Notes.**
s.d.,  standard deviation.

**Table 3  Quantitative characteristics of the category features of the Takashi2019 diabetes type 1 dataset.**

| Category feature | # | % |
|---|---|---|
| Insulin regimen (0: CSII) | 39 | 53.42 |
| Insulin regimen (1: MDI) | 34 | 46.58 |
| Sex (0: female) | 51 | 69.87 |
| Sex (1: male) | 22 | 30.13 |
| Total | 73 | 100.00 |

**Notes.**
[#] Number of patients at the medical check-up.
[%] Percentage of of patients at the medical check-up.

## Diabetes type 2 dataset

The AlOlaiwi2018 diabetes type 2 dataset contains data of 400 patients from Saudi Arabia (*AlOlaiwi, AlHarbi & Tourkmani, 2018*). Each patient profile has 49 clinical features, including one indicating the past duration of diabetes type 2.

The original dataset curators *AlOlaiwi, AlHarbi & Tourkmani (2018)* collected these data at the Alwazarat Health Care Center (Riyadh, Saudi Arabia) from 1st April 2017 to 20th March 2018.

The AlOlaiwi2018 diabetes type 2 dataset consists of several features related to conditions of the patient (diabetic retinopathy, bloating, postural heart rate, vomiting, stomach fullness, belly visibly larger, gastroparesis, hypertension), physiological traits (sex, age,

body-mass index), treatment (metformin, insulin, sulfonylurea), variables related to lifestyle (smoking). and laboratory test results features (eGFR, cholesterol, tryglycerides, albumn-to-creatinine ratio, hemogloblin) (Table 4).

This diabetes type 2 dataset contains data of patients 55.25 years old on average, with 56.25% women and 43.75% men (Tables 5 and 6).

The duration of diabetes type 1 for the Takashi2019 diabetes type 1 dataset patients is 25.68 years on average, and ranges between 10 and 41 years (Fig. 1). For the diabetes 2 patients of the AlOaiwi2018 dataset, instead, the duration of diabetes is 10.77 years on average, with values that range between 0.1 and 30 years (Fig. 1).

The two datasets share seven common features: age, eGFR, HbA1c, insulin intake, sex, body-mass index, and of course diabetes past duration. Additional information about the two datasets is available in the original publications (*Takashi et al., 2019*; *AlOlaiwi, AlHarbi & Tourkmani, 2018*).

## METHODS

To predict the past diabetes duration for each dataset, we made a regression analysis employing several machine learning methods: Random Forests (*Breiman, 2001*), XGBoost (*Chen & Guestrin, 2016*), Linear Regression (*Groß, 2012*), Decision Trees (*Quinlan, 1990*).

We chose these data mining algorithms because they showed their strength in several biomedical informatics studies involving electronic health records in the past (*Chicco & Jurman, 2020b*; *Chicco et al. 2023*; *Cerono, Melaiu & Chicco, 2023*), including studies of DREAM Challenges (*Meyer & Saez-Rodriguez, 2021*). Moreover, tree-based machine learning algorithms are especially suitable for medical data, because they can help physicians decision-making (*Podgorelec et al., 2002*).

Both datasets had missing values. We addressed this problem by using the algorithm Multivariate Imputation by Chained Equations (MICE) (*van Buuren & Groothuis-Oudshoorn, 2010*) of the known Python package `scikit-learn` (*Buitinck et al., 2013*), under the assumptions that these values were missing at random. The MICE algorithm imputes missing data through an iterative series of predictive models utilizing other variables in the dataset.

We employed machine Learning regression algorithms directly from `scikit-learn`, utilizing the default values from the library for the multiple parameters available. For the regression analysis, we ran 1,000 executions with 70% randomly chosen elements for the training set and the remaining 30% for the test set (*Chicco, 2017*), both for regression and feature ranking through recursive feature elimination (RFE) (*Darst, Malecki & Engelman, 2018*).

For the diabetes past duration prediction, we employed all the variables and then saved the results measured with traditional regression rates such as the coefficient of determination ($R^2$), root mean square error (RMSE), mean square error (MSE), mean absolute error (MAE), and symmetric mean absolute percentage error (SMAPE). We reported their formulas in the Supplementary Information.

For the recursive feature elimination, we repeated the tests for the numbers of features, by eliminating one feature at each run. Here we only used Random Forests, because it is

**Table 4** Meaning and measurement unit of the variables of the AlOlaiwi2018 diabetes type 2 dataset.

| Feature name | Measurement | Meaning |
|---|---|---|
| Age | Years | Age of the patient at the medical consult |
| Albuminuria | Categories | Normoalbuminaria: 0, microalbuminuria: 1, macroalbuminuria: 2 |
| Anti HTN | Binary | Taking any hipertensive drugs. 0: No 1: Yes |
| Bloating | Binary | Patient suffering from bloating: No: 0, Yes: 1 |
| BMI | kg/m*2 | Body mass index |
| CAN | Binary | Patient suffering from cardiovascular autonomic neuropathy. No: 0, Yes: 1 |
| DBP | mmHg | Diastolic blood pressure |
| DDP-4 inhibitor | Binary | Prescribed DPP4 inhibitor. 0: No 1: Yes |
| DR | Binary | Diabetic retinopathy. 0: No, 1: Yes. |
| Duration of DM | Years | Duration of diabetes mellitus type 2 in years |
| eGFR MDRD equation | ml/min | Estimated glomerular filtration rate by the MDRD study equation |
| Excessive fullness after meals | Binary | Patient suffering from excessive fullness after meals: No: 0, Yes: 1 |
| FBS | mmol/L | Fasting Blood Sugar. |
| GCSI category | Category | Gastroparesis cardinal sympation index, Classified as categories: None: 0, Mild: 1, Severe: 2. |
| GCSI new | Point Scores | Gastroparesis cardinal sympton index score. |
| GCSI present ? | Binary | Gastroparesis symptomps: absent: 0, present: 1 |
| GCSI score | Point scores | Gastroparesis cardinal symptom index score. |
| HbA1c | % | Percetange of glycosylated hemoglobin |
| HDL | mmol/L | High density lipoprotein |
| HTN | Binary | Hypertension: 0: No 1: Yes |
| Insulin | Binary | Taking insulin: 0: No 1: Yes |
| LDL | mmol/L | Low-density lipoprotein |
| Loss of appetitie | Binary | Loss of appetite for the last 2 weeks. No: 0, Yes: 1 |
| Meglitinides | Binary | Use of Meglitinides. 0: No 1: Yes |
| Metformin | Binary | Use of metformin. 0: No 1: Yes |
| Nausea | Binary | Feelings of nausea in the last 2 weeks. No: 0, Yes: 1 |
| None | Binary | Not taking any drug at all? 0: No 1: Yes |
| Not able to finish a meal | Binary | Inability to finish a regular size meal. No: 0, Yes: 1 |
| Orthostatic hypotension | Binary | Patients suffering from orthostatic hypotension: No: 0, Yes: 1 |
| PDBP | mmHg | Diastolic blood pressure after postural manoeuvres. |
| PHR | bpm | Postural heart rate |
| Presence of any symptom | Binary | Presence of any gastroparesis symptom: No: 0, Yes: 1 |
| PSBP | mmHg | Systolic blood pressure after postural manoeuvres |
| QTc | Seconds | Corrected QT interval. (measured in the EKG) |
| QTc prolonged | Category | Corrected QT interval prolongation: No: 0, Borderline: 0.5 Yes: 1 |

**Table 4** (*continued*)

| Feature name | Measurement | Meaning |
|---|---|---|
| Resting tachycardia | Binary | Patient suffering from resting tachycardia: No: 0, Yes: 1 |
| Retching | Binary | Patient suffering from retching: No: 0, Yes: 1 |
| SBP | mmHg | Systolic blood pressure |
| Sex | Binary | Patient's sex: 0: female, 1: male |
| Smoking | Binary | Patient smoking habit: 0: No, 1: Yes |
| Stomach fullness | Binary | Patient suffering from stomach fullness: No: 0, Yes: 1 |
| Stomach or belly visibly larger | Binary | Patient suffering from belly visibly larger: No: 0, Yes: 1 |
| Sulfonylurea | Binary | Patient using sulfonylurea: 0: No 1: Yes |
| TC | mmol/L | Total cholesterol |
| TG | mmol/L | Triglycerides |
| TZD | Binary | Patient using thiazolidinediones: 0: No 1: Yes |
| UACR new | mg/g | Urine albumin-to-creatinine ratio |
| Urine ACR | mg/g | Urine albumin to creatinine ratio 6 months before. |
| Vomiting | Binary | Patient suffering from Vomiting: No: 0, Yes: 1 |

**Notes.**
$^{kg/m*2}$kilogram per meter squared.
$^{mmHg}$millimeters of Mercury.
$^{ml/min}$milliliters per minutes.
$^{mmol/L}$millimole per liter.
$^{bpm}$beats per minutes.
$^{mg/g}$urine Albumin (mg/dL) / urine creatinine (g/dL).

the method which achieved the higher R-squared in the past diabetes duration prediction. We computed and saved the coefficient of determination for each test, and generated the ranking of the dataset features based on the increasing value of R-squared: the lower the R-squared when a specific feature is removed, the more important that feature is *Chicco, Warrens & Jurman (2021)*. We repeated these tests 1,000 times and then merged the final rankings with the Borda's method (*Lansdowne & Woodward, 1996*). The Borda's count method consist in adding up the ranks of each variable for each iteration, resulting in a single fused ranking score after the 1,000 iterations.

This ensemble machine learning approach generated a standing of variables from tests where the features interact between each other. To further verify the importance of the datasets variables, we also produced a biostatistics ranking based on a traditional univariate test, the Kruskal–Wallis test (*Kruskal & Wallis, 1952*; *McKight & Najab, 2010*). The Kruskal–Wallis test applied to two numerical vectors of the same size generates $p$-values in the $[0, 1]$ interval: if the two vectors are correlated, the test $p$-value is close to 0; on the contrary, if there is no correlation between the two vectors, the resulting $p$-value is close to 1. We performed this operation to see how each feature alone relates to the past diabetes duration, without interference from the other clinical variables. Following the recent biostatistics guidelines by *Benjamin et al. (2018)* we considered significant only the variables that obtained a $p$-value lower than 0.005, differently from 0.05 as traditionally done in the past.

**Table 5** Quantitative characteristics of the category features of the AlOlaiwi2018 diabetes type 2 dataset.

| category feature | # | % |
|---|---|---|
| albuminuria: macroalbuminuria | 18 | 4.50 |
| albuminuria: microalbuminuria | 84 | 21.00 |
| albuminuria: normoalbuminuria | 298 | 74.50 |
| anti HTN: no | 143 | 35.75 |
| anti HTN: yes | 257 | 64.25 |
| bloating: no | 225 | 56.25 |
| bloating: yes | 175 | 43.75 |
| CAN: no | 339 | 84.75 |
| CAN: yes | 61 | 15.25 |
| DDP-4 inhibitor: no | 247 | 61.75 |
| DDP-4 inhibitor: yes | 153 | 38.25 |
| DR: no | 254 | 63.50 |
| DR: yes | 77 | 36.50 |
| excessive fullness after meals: no | 265 | 66.25 |
| excessive fullness after meals: yes | 135 | 33.75 |
| GCSI category: mild | 256 | 64.00 |
| GCSI category: none | 143 | 35.75 |
| GCSI category: severe | 1 | 0.25 |
| GCSI present: absent | 375 | 93.75 |
| GCSI present: present | 25 | 6.25 |
| HTN: no | 239 | 59.75 |
| HTN: yes | 161 | 40.25 |
| Insulin: no | 211 | 52.75 |
| Insulin: yes | 189 | 47.25 |
| loss of appetitie: no | 305 | 76.25 |
| loss of appetitie: yes | 95 | 23.75 |
| meglitinides: no | 399 | 99.75 |
| meglitinides: yes | 1 | 0.25 |
| metformin: no | 22 | 5.50 |
| metformin: yes | 378 | 94.50 |
| nausea: no | 327 | 81.75 |
| nausea: yes | 73 | 18.25 |
| none: no | 398 | 99.50 |
| none: yes | 2 | 0.50 |
| not able to finish a meal: no | 261 | 75.25 |
| not able to finish a meal: yes | 139 | 34.75 |
| orthostatic hypothension: no | 388 | 97.00 |
| orthostatic hypothension: yes | 12 | 3.00 |
| QTc prolonged: borderline | 122 | 30.50 |
| QTc prolonged: no | 247 | 61.75 |
| QTc prolonged: yes | 31 | 7.75 |

**Table 5** (*continued*)

| category feature | # | % |
|---|---|---|
| resting tachycardia: no | 377 | 94.25 |
| resting tachycardia: yes | 23 | 5.75 |
| retching: No | 357 | 89.25 |
| retching: Yes | 43 | 10.75 |
| sex Female | 225 | 56.25 |
| sex Male | 175 | 43.75 |
| smoking 0: no | 359 | 89.75 |
| smoking 1: yes | 41 | 11.25 |
| stomach fullness: no | 273 | 68.25 |
| stomach fullness: yes | 127 | 31.75 |
| stomach or belly visibly larger: no | 286 | 71.25 |
| stomach or belly visibly larger: yes | 114 | 28.75 |
| sulfonylurea: no | 202 | 50.50 |
| sulfonylurea: yes | 198 | 49.50 |
| TZD: no | 397 | 99.25 |
| TZD: yes | 3 | 0.75 |
| vomiting: no | 383 | 95.75 |
| vomiting: yes | 17 | 4.25 |
| total | 400 | 100% |

**Notes.**
[#]Number of patients at the medical check-up.
[%]Percentage of the patients at the medical check-up.

# RESULTS

In this section, we first report and describe the results obtained by the regression analysis for the prediction of the past diabetes duration ('Prediction of the past diabetes duration'), and then we report and describe the results obtained by regression methods and biostatistics for feature ranking ('Clinical feature ranking results').

## Prediction of the past diabetes duration

Among the four machine learning algorithms employed for regression, Random Forests outperformed the other three methods on both the datasets, achieving an average coefficient of determination of +0.41 on the Takashi2019 diabetes type 1 dataset and an average coefficient of determination of +0.35 on the AlOlaiwi2018 diabetes type 2 dataset (Tables 7 and 8).

On the diabetes type 1 dataset, Random Forests obtained the top R-squared, root mean square error, and mean square error, but was outperformed by XGBoost on the mean absolute error and on the symmetric mean absolute percentage error (Table 7). The two regression analyses generated the same standings for the results based on R-squared: Random Forests on first position, then XGBoost followed by Linear Regression, with Decision Trees on the last position (Fig. 2).

The scatterplots of the top performing methods (Fig. 3) shows that the majority of points is close to the $x = y$ line, which corresponds to perfect prediction.

**Table 6** Quantitative characteristics of the numeric features of the AlOlaiwi2018 diabetes type 2 dataset.

| Numeric feature | Median | Mean | s.d. | Range |
|---|---|---|---|---|
| Age | 55 | 55.25 | 10.646 | [28, 85] |
| BMI | 32 | 32.46 | 5.40 | [17.6, 48] |
| DBP | 74 | 74.52 | 9.52 | [42, 105] |
| Duration of diabetes [target] | 10 | 10.77 | 6.89 | [0.1, 30] |
| eGFR MDRD equation | 100.35 | 102.02 | 25.10 | [42.1, 183.1] |
| FBS | 7.7 | 8.71 | 3.55 | [3.1, 25.6] |
| GCSI new | 0.4 | 0.65 | 0.67 | [0, 3.2] |
| GCSI score | 4 | 5.95 | 6.04 | [0, 29] |
| HbA1c | 7.7 | 8.07 | 1.59 | [4.8, 15] |
| HDL | 1.12 | 1.15 | 0.34 | [0.38, 3.23] |
| LDL | 2.41 | 2.55 | 0.78 | [0.99, 6.3] |
| PDBP | 79 | 79.47 | 9.06 | [55, 110] |
| PHR | 90 | 79.78 | 13.19 | [48, 136] |
| PSBP | 132 | 133.95 | 16.09 | [99, 189] |
| QTc | 0.43 | 0.43 | 0.03 | [0.36, 0.6] |
| SBP | 130 | 103.32 | 17.08 | [11, 195] |
| TC | 4.04 | 4.19 | 0.89 | [1.81, 7.96] |
| TG | 1.52 | 1.70 | 0.81 | [0.3, 7.17] |
| UACR new | 9.155 | 59.92 | 194.49 | [1.14, 2103] |
| Urine ACR | 1.05 | 6.82 | 22.00 | [0.16, 237.9] |

**Notes.**

s.d., standard deviation.

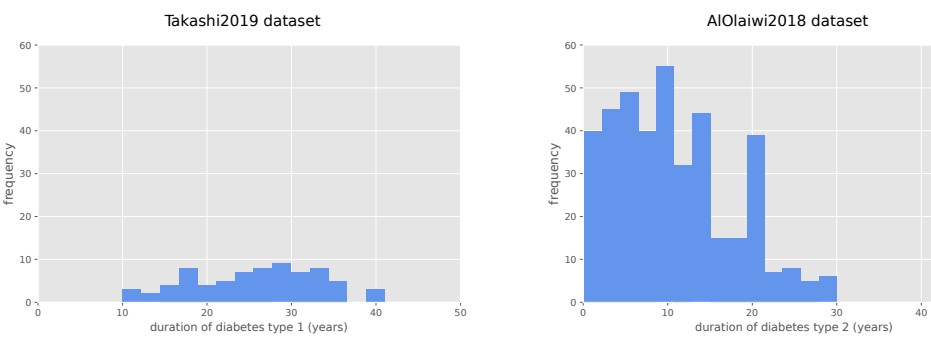

**Figure 1** **Frequency histograms of diabetes duration.** Duration of diabetes type 1 for the Takashi2019 dataset (left) and diabetes type 2 for the AlOlaiwi2018 dataset (right).

Regarding SMAPE, XGBoost obtained the top result of 0.21, corresponding to 89.5% correctness in the $[0, 2]$ interval, on the diabetes type 1 dataset. Random Forests achieved the top SMAPE score of 0.47 on the diabetes type 2 dataset (Table 7), which corresponds to 76.5% correctness in the same interval (Table 8). Decision Trees obtained poor results on both dataset: an average coefficient of determination close to zero ($R^2 = 0.05$) in
**Table 7   Regression results for the prediction of the duration of diabetes type 1 on the Takashi2019 dataset.** Performance of the learned models with the different methods evaluated with the different metrics, expressed in the format "average value $\pm$ standard deviation", obtained on 1,000 executions, each execution had 70% randomly chosen data instances for training set and the remaining 30% used for test set. We reported in blue and with an asterisk * the top result for each rate. At the beginning of each execution we randomly shuffled the dataset instances. RMSE: root mean square error. MAE: mean absolute error. MSE: mean square error. SMAPE: symmetric mean absolute percentage error. $R^2$: coefficient of determination. RMSE, MAE, MSE: best value 0 and worst value $+\infty$. $R^2$: best value $+1$ and worst value $-\infty$. SMAPE: best value 0 and worst value 2. We listed the complete formulas of $R^2$, RMSE, MSE, MAE, and SMAPE in the Supplemental Information. We ranked the methods considering the results obtained through R-squared (in bold).

| Method | $R^2$ | RMSE | MAE | MSE | SMAPE |
|---|---|---|---|---|---|
| Random forests | *0.41 $\pm$ 0.05 | *5.98 $\pm$ 0.27 | 5.19 $\pm$ 0.26 | *35.87 $\pm$ 03.31 | 0.22 $\pm$ 0.01 |
| XGBoost | 0.39 $\pm$ 0.14 | 6.04 $\pm$ 0.70 | *5.00 $\pm$ 0.49 | 37.08 $\pm$ 08.97 | *0.21 $\pm$ 0.02 |
| Linear regression | 0.14 $\pm$ 0.47 | 7.00 $\pm$ 1.83 | 5.52 $\pm$ 1.31 | 52.49 $\pm$ 29.27 | 0.27 $\pm$ 0.06 |
| Decision trees | 0.05 $\pm$ 0.26 | 7.53 $\pm$ 1.07 | 6.23 $\pm$ 0.88 | 57.98 $\pm$ 16.46 | 0.26 $\pm$ 0.03 |

**Table 8   Regression results for the prediction of the duration of diabetes type 2 on the AlOlaiwi2018 dataset.** These results refer to the same abbreviation meanings and execution details of Table 7 caption.

| Method | $R^2$ | RMSE | MAE | MSE | SMAPE |
|---|---|---|---|---|---|
| Random forests | *0.35 $\pm$ 0.02 | *5.64 $\pm$ 0.11 | *4.60 $\pm$ 0.10 | *31.85 $\pm$ 1.30 | *0.47 $\pm$ 0.01 |
| XGBoost | 0.25 $\pm$ 0.06 | 6.07 $\pm$ 0.24 | 4.67 $\pm$ 0.21 | 36.91 $\pm$ 2.98 | 0.49 $\pm$ 0.02 |
| Linear regression | 0.09 $\pm$ 0.07 | 6.67 $\pm$ 0.27 | 5.18 $\pm$ 0.21 | 44.54 $\pm$ 3.67 | 0.52 $\pm$ 0.02 |
| Decision trees | $-$0.21 $\pm$ 0.15 | 7.71 $\pm$ 0.47 | 5.98 $\pm$ 0.39 | 59.69 $\pm$ 7.32 | 0.61 $\pm$ 0.04 |

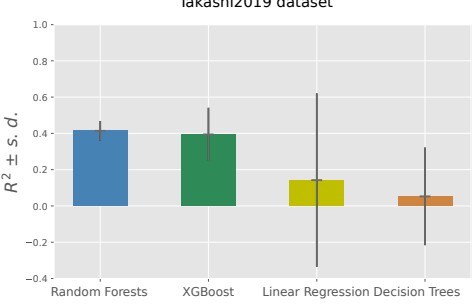
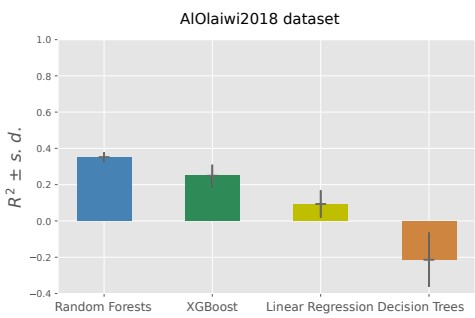

**Figure 2   Regression results on the Takashi2019 diabetes type 1 dataset (left) and on the AlOlaiwi2018 diabetes type 2 dataset (right).** Representation of the Regression results reported as mean coefficient of determination $\pm$ the corresponding standard deviations for each method. We reported the complete results measured with other rates in Tables 7 and 8.

Takashi2019 diabetes type 1 dataset and a negative average coefficient of determination ($R^2 = -0.21$) in the AlOlaiwi2018 diabetes type 2 dataset.

## Clinical feature ranking results

The feature ranking phase based on Random Forests and recursive feature elimination (RFE) generated a standing of the datasets variables, sorted by predictive importance. On

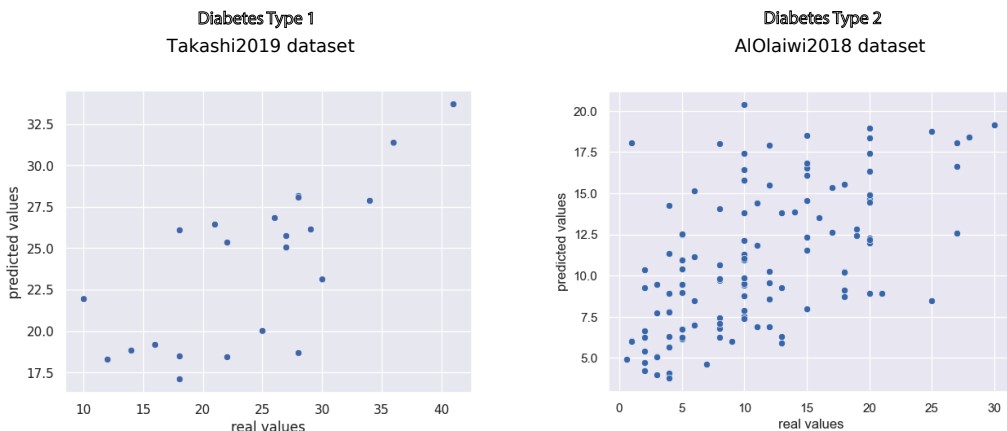

**Figure 3** **Scatterplot of the prediction results of the top methods on the Takashi2019 diabetes type 1 dataset (left) and on the AlOlaiwi2018 diabetes type 2 dataset (right).** Representation of the regression results reported as actual real values *versus* predicted values, obtained through the top methods. We reported the complete results measured with other rates in Tables 7 and 8.

the Takashi2019 diabetes type 1 dataset, the key variables for the prediction of past diabetes duration resulted being age, daily bolus dose of insulin, and gait speed (Table 9). Among the most important variables, we also noticed estimated glomerular filtration rate (eGRF), total daily dose of insulin, grip strength, and body-mass index (BMI) (Table 9). On the bottom of the standing, the RFE put the weight of the patient, the insulin regimen, and the level of undercarboxilated osteocalcin (Table 9).

On the same Takashi2019 dataset, we also computed the feature ranking by using a traditional univariate statistics method: the Kruskal–Wallis test (*McKight & Najab, 2010*). We computed this test between each variable and the target variable (duration of diabetes type 1), and ranked the resulting *p*-values in increasing order. The results showed that no clinical variable obtained a *p*-value lower than 0.005, so no feature resulted being significant in relation with the past duration of diabetes type 1 (Table S1).

Regarding the diabetes type 2 dataset of AlOlaiwi2018, the ensemble machine learning recursive feature elimination indicated diabetic retinopathy (DR), age, insulin intake, body-mass-index, and diastolic blood pressure after postural manoeuvres (PDBP) as the top five most predictive variables for past duration of diabetes type 2 (Table 10). The same ranking indicated nausea, eGFR, and the inability to finish as the least predictive variables in the dataset (Table 10).

The biostatistics feature ranking based on the univariate Kruskal–Wallis test found nine significant variables, which obtained *p*-values lower than the 0.005 threshold: if the patients takes no drug at all, age, insulin, diastolic blood pressure after postural manoeuvres (PDBP), diastolic blood pressure (DBP), if the patient takes in thiazolidinediones (TZD), diastolic blood pressure (PDBP), if the patient takes in metformin, and if the patient takes in sulfonylurea (Table S2). The feature indicating if the patient takes no drugs at all (none), in particular, obtained a *p*-value much lower than the other variables ($2.77 \times 10^{-27}$), which highlights its importance in the dataset.

**Table 9** **Feature ranking results obtained through Random Forests on the Takashi2019 diabetes type 1 dataset.** We computed the average Borda score on 1,000 executions of Random Forests. At the beginning of each execution we randomly shuffled the dataset instances.

| Rank | Feature | Average borda score | s.d. |
|---|---|---|---|
| 1 | Age | 1.275 | 1.719 |
| 2 | Bolus | 6.955 | 5.488 |
| 3 | Gait speed | 7.537 | 5.657 |
| 4 | eGFR | 8.244 | 5.683 |
| 5 | TDD | 9.683 | 5.467 |
| 6 | Grip strength | 10.517 | 5.116 |
| 7 | BMI | 10.578 | 5.319 |
| 8 | Adiponectin | 10.991 | 5.019 |
| 9 | Basal | 11.025 | 4.959 |
| 10 | HbA1c | 11.063 | 4.906 |
| 11 | Bodyfat | 11.139 | 4.664 |
| 12 | OC | 11.177 | 4.536 |
| 13 | Sex | 11.178 | 5.174 |
| 14 | Free-test | 11.200 | 4.798 |
| 15 | Knee extension strength | 11.322 | 4.777 |
| 16 | SMI | 11.458 | 4.521 |
| 17 | ucOC | 11.459 | 4.497 |
| 18 | Insulin regimen | 11.564 | 4.809 |
| 19 | Added weight | 11.635 | 4.535 |

**Notes.**
s.d., standard deviation.

# DISCUSSION

In this section, we discuss the results we obtained in our scientific analyses, report some key take-home messages inferred in this study, and describe some limitations and potential future development.

## Prediction of past duration of diabetes

Our regression results on the two datasets proof that ensemble machine learning can efficiently predict the past duration of diabetes from the electronic health records of patients. The fact that our computational intelligence methods were able to obtain good results not only on one dataset but also on a second one confirms the efficacy of our approach, both on diabetes type 1 and on diabetes type 2. The Random Forests method, in particular, obtained the top results measured with the coefficient of determination both on the diabetes type 1 Takashi2019 dataset and on the diabetes type 2 AlOlaiwi2018 dataset. The gradient boosting method XGBoost, also, achieved good prediction results on both the datasets, while Linear Regression and Decision Trees did not.

These results confirm the effectiveness of ensemble machine learning and, in particular, of the Random Forests method in health informatics. Random Forests, in fact, resulted being the top performing method in multiple previous studies in this field (*Chicco & Rovelli, 2019*; *Chicco & Jurman, 2021*; *Chicco & Jurman, 2020a*).

**Table 10 Feature ranking results obtained through Random Forests on the AlOlaiwi2018 diabetes type 2 dataset.** We computed the average Borda score on 1,000 executions of Random Forests. At the beginning of each execution we randomly shuffled the dataset instances.

| Rank | Feature | Average borda score | s.d. |
|------|---------|---------------------|------|
| 1 | DR | 3.907 | 8.451 |
| 2 | Age | 6.133 | 10.186 |
| 3 | Insulin | 6.843 | 9.626 |
| 4 | BMI | 17.256 | 14.731 |
| 5 | PDBP | 18.721 | 14.604 |
| 6 | CAN | 22.524 | 13.719 |
| 7 | Sulfonylurea | 23.417 | 14.038 |
| 8 | HDL | 23.645 | 13.851 |
| 9 | FBS | 23.927 | 13.910 |
| 10 | LDL | 24.437 | 13.546 |
| 11 | Anti HTN | 24.524 | 13.439 |
| 12 | SBP | 24.533 | 13.825 |
| 13 | DDP-4 inhibitor | 24.581 | 13.244 |
| 14 | PHR | 25.162 | 13.187 |
| 15 | Urine ACR | 25.177 | 12.435 |
| 16 | DBP | 25.241 | 13.712 |
| 17 | QTc | 25.412 | 13.239 |
| 18 | TC | 25.465 | 13.568 |
| 19 | TG | 25.681 | 13.450 |
| 20 | HbA1c | 25.742 | 12.959 |
| 21 | Sex | 25.959 | 12.897 |
| 22 | GCSI present ? | 26.133 | 12.506 |
| 23 | UACR new | 26.176 | 12.751 |
| 24 | HTN | 26.342 | 12.838 |
| 25 | Metformin | 26.542 | 12.599 |
| 26 | PSBP | 26.619 | 12.986 |
| 27 | Resting tachycardia | 26.623 | 12.656 |
| 28 | GCSI score | 26.641 | 12.466 |
| 29 | Excessive fullness after meals | 26.686 | 12.530 |
| 30 | Vomiting | 26.723 | 12.413 |
| 31 | Meglitinides | 26.830 | 12.307 |
| 32 | Albuminuria | 26.849 | 12.262 |
| 33 | Loss of appetitie | 26.850 | 12.532 |
| 34 | Bloating | 26.858 | 12.994 |
| 35 | TZD | 26.915 | 12.335 |
| 36 | Retching | 26.947 | 12.862 |
| 37 | Stomach fullness | 26.976 | 12.787 |
| 38 | Orthostatic hypotension | 27.144 | 12.411 |
| 39 | GCSI new | 27.215 | 12.696 |

**Table 10** (*continued*)

| Rank | Feature | Average borda score | s.d. |
|------|---------|---------------------|------|
| 40 | Stomach or belly visibly larger | 27.235 | 12.439 |
| 41 | Smoking | 27.242 | 12.754 |
| 42 | Presence of any symptom | 27.304 | 12.328 |
| 43 | None | 27.317 | 12.805 |
| 44 | QTc prolonged | 27.332 | 12.307 |
| 45 | GCSI category | 27.476 | 13.003 |
| 46 | Nausea | 27.546 | 12.334 |
| 47 | eGFR MDRD equation | 27.557 | 12.833 |
| 48 | Not able to finish a meal | 27.635 | 12.170 |

**Notes.**
s.d., standard deviation.

Medical evidence from the scientific literature confirm the importance of diabetes past duration. Patients with long standing diabetes type 2, in fact, might have troubles controlling their glycemia (*Hayashino et al., 2017*). Additionally, patients who suffered diabetes for a longer time often are more in need of receiving insulin treatments, for obvious reasons (*Duckworth et al., 2011*).

Revealing the duration of diabetes therefore can help with the establishment of a better therapy, since a longer duration of this disease has been linked with poor glycemic control and with the consequent need of more complex medical treatment. Moreover, researchers also recorded an increase in risk of ischemic stroke in correlation with a long diabetes duration (*Banerjee et al., 2012*).

## Feature ranking for past duration of diabetes

As mentioned earlier ('Datasets'), the two datasets share six common variables, in addition to past diabetes duration. Age resulted being the top most important variable in the Takashi2019 diabetes type 1 dataset feature ranking and the second most important factor in the AlOaiwi2018 dataset standing ('Clinical feature ranking results'). This result comes with no surprise: in the medical community it is known that age is proportional to the duration of both diabetes type 1 and type 2 (*Wannamethee et al., 2011*; *Zoungas et al., 2014*).

Expectedly, insulin obtained a high ranking position on both standings (*Davies et al., 2013*). In the diabetes 1 dataset, the daily bolus dose of insulin taken by the patients was ranked second most important factor, while in the diabetes 2 dataset the information about the patient taking insulin or not was ranked top most relevant feature ('Clinical feature ranking results').

An interesting aspect of both rankings came from the positions of body-mass index in the two standings. Both the feature rankings, in fact, listed body-mass index as a top most important factor: it is found on the 7th position of the Takashi2019 diabetes type 1 dataset standing and on the 4th position of the AlOlaiwi2018 diabetes type 2 dataset standing. Several studies confirm the association between body-mass index and duration of diabetes (*Bray et al., 2008*; *Funakoshi et al., 2008*; *Pencek et al., 2012*).

Both the feature rankings gave average importance to HbA1c (10th position on the Takashi2019 diabetes type 1 dataset ranking and 20th position on the AlOlaiwi2018 diabetes type 2 dataset ranking), while they gave a discordant outcome for the eGFR (top position on the diabetes type 1 ranking and low position for the diabetes type 2 ranking); HbA1c is known to have an association with diabetes (*Sherwani et al., 2016*). Both standings listed sex as unimportant variable (13th position on the Takashi2019 diabetes type 1 dataset ranking and 21th position on the AlOlaiwi2018 diabetes type 2 dataset ranking).

These results confirm the importance of age, insulin intake, and body-mass index in the prediction of diabetes past duration from electronic health records. The role of body-mass index, especially, comes of great importance: our study results suggest that physicians and medical doctors can focus on this clinical factor to predict the past duration of diabetes, when this information is unavailable. Medical doctors can then take advantage of this inferred information for clinical decision-making, that is to decide which treatment for the patient, which screening tests, which medicines to prescribe, and all the other details.

## CONCLUSIONS

Knowing the how long a patient had diabetes is a critical information for the medical doctors to establish the correct treatment. Different durations, in fact, require different screenings, medicines, and therapies.

Even if pivotal, this information might be unavailable for patients, especially if they have just been diagnosed: since the diabetes type 2 can appear without symptoms, the diabetes diagnosis sometimes can arrive years or even decades after the diabetes onset. In these cases, a method that can calculate the past duration of diabetes in a patient from her/his clinical records can be extremely useful.

In this study, we applied several computational intelligence methods on two datasets of electronic health records of patients with diabetes (a dataset of T1DM and a dataset of T2DM) for this scope. On both the datasets, our machine learning models were able to efficiently predict the past duration of diabetes, obtaining a top average $R^2 = 0.41$ on the Takashi2019 diabetes type 1 dataset and a top average $R^2 = 0.35$ on the AlOaiwi2018 dataset.

After verifying the predictive efficacy of our machine learning methods for this task, we computed the feature rankings of these two datasets, through a traditional recursive feature elimination procedure. The feature ranking phase indicated age, insulin, and body-mass index as most important predictive factors on both the datasets, suggesting therefore physicians and medical doctors to focus on these elements of clinical records to foresee the duration of diabetes for any possible patient. To the best of our knowledge, no previous study utilized computational intelligence to forecast past diabetes duration and to detect the most relevant predictive variables for this scope.

Diabetic patients have increased risk of suffering from multiple and diverse diseases. Strict screening looking for early signs of pathogenesis depending on age of patients and duration of diabetes can be very useful for a correct diagnosis and prognosis. Regular diabetes type 1 usually has a sudden clinical presentation, so duration of disease is often

known, but for diabetes type 2 and LADA (Latent Autoimmune Diabetes in Adults) sub variation of diabetes type 1 (*Pieralice & Pozzilli, 2018*; *Isomaa et al., 1999*), the presentation is slow and often goes misdiagnosed for years. In this context, our machine learning approach could be an effective way to retrospectively predict duration from onset.

Our computational models would allow doctors to start screening LADA patients at the right time. For example, type 1 diabetic patients generally do not develop retinopathy within 3–5 years from the diagnosis, we start screening for it with a fundoscopy after 3 years from diagnosis (*Fong et al., 2004*) A patient with LADA could be diagnosed 2 years late from the actual start of the disease, and therefore be 2 years late for screening as we would falsely assign a later onset.

As a limitation, we have to report that it would have been useful to have additional diabetes datasets where to verify our findings. We found other studies about analyses on electronic health records of patients with diabetes (*Bächle et al., 2015*; *Al-Rubeaan et al., 2015*; *Zabeen et al., 2016*; *Moser et al., 2018*); we contacted the corresponding authors of each of them and requested the datasets, but received no reply or our requests were rejected.

In the future, we plan to further investigate diabetes duration by analyzing data of other sources and types, such as microarray gene expression (*Choi et al., 2008*), RNA-Seq gene expression (*Rubin et al., 2016*), medical images (*Samant & Agarwal, 2018*), and others. We also plan to investigate data of other diseases such as heart failure (*Shin et al., 2021*) and amyotrophic lateral sclerosis (*Kueffner et al., 2019*).

**Abbreviations**

| | |
|---|---|
| **ACR** | albumin to creatinine ratio |
| **BMI** | body-mass index |
| **CAN** | cardiovascular autonomic neuropathy |
| **CSII** | continuous subcutaneous injections |
| **DBP** | diastolic blood pressure |
| **DDP-4** | dipeptidyl peptidase-4 inhibitor |
| **T1DM** | diabetes mellitus type 1 |
| **T2DM** | diabetes mellitus type 2 |
| **DR** | diabetic retinopathy |
| **eGFR** | estimated glomerular filtration rate |
| **EHRs** | electronic health records |
| **FBS** | fasting blood glucose |
| **GCSI** | gastroparesis cardinal symption index |
| **HbA1C** | percetange of glycosylated hemoglobin |
| **HDL** | high density lipoprotein |
| **HTN** | hypertension |
| **LDL** | low-density lipoprotein |
| **MAE** | mean absolute error |
| **MDI** | multiple daily injections |
| **MSE** | mean squared error |
| **OC** | osteocalcin |
| **PDBP** | postural diastolic blood pressure |

| | |
|---|---|
| **PHR** | postural heart rate |
| **PSBP** | postural systolic blood pressure |
| **QTc** | corrected QT interval |
| **RFE** | recursive feature elimination |
| **RMSE** | root mean square error |
| **SBP** | systolic blood pressure |
| **SMAPE** | symmetric mean absolute percentage error |
| **SMI** | skeletal muscle mass index |
| **TC** | total cholesterol |
| **TG** | triglycerides |
| **T1D** | diabetes type 1 |
| **T2D** | diabetes type 2 |
| **TDD** | total daily dose |
| **TZD** | thiazolidinediones |
| **UACR** | urine albumin to creatinine ratio |
| **ucOC** | undercarboxylated osteocalcin |

## ACKNOWLEDGEMENTS

The authors would like to thank Javier Nunez and Cecilia Andrea Spinelli for their useful suggestions.

### Funding

This study was funded by the European Union–Next Generation EU programme, in the context of The National Recovery and Resilience Plan, Investment Partenariato Esteso PE8 "Conseguenze e sfide dell'invecchiamento", Project Age-It (Ageing Well in an Ageing Society), and was supported by Ministero dell'Università e della Ricerca of Italy under the "Dipartimenti di Eccellenza 2023-2027" ReGAInS grant assigned to Dipartimento di Informatica Sistemistica e Comunicazione at Università di Milano-Bicocca. The funders had no role in study design, data collection and analysis, decision to publish, or preparation of the manuscript.

### Grant Disclosures

The following grant information was disclosed by the authors:
The European Union–Next Generation EU programme, in the context of The National Recovery and Resilience Plan, Investment Partenariato Esteso PE8 "Conseguenze e sfide dell'invecchiamento", Project Age-It (Ageing Well in an Ageing Society).
Ministero dell'Università e della Ricerca of Italy under the "Dipartimenti di Eccellenza 2023-2027" ReGAInS.
Dipartimento di Informatica Sistemistica e Comunicazione at Università di Milano-Bicocca.

## Competing Interests

Davide Chicco is an Academic Editor for PeerJ Computer Science.

## Author Contributions

- Gabriel Cerono conceived and designed the experiments, performed the experiments, analyzed the data, performed the computation work, prepared figures and/or tables, and approved the final draft.
- Davide Chicco conceived and designed the experiments, analyzed the data, authored or reviewed drafts of the article, and approved the final draft.

## Data Availability

The Takashi2019 diabetes type 1 dataset was collected at the Osaka University Hospital and Osaka Police Hospital (Osaka, Japan) and is available at figshare: Takashi, Yuichi; Ishizu, Masashi; Mori, Hiroyasu; Miyashita, Kazuyuki; Sakamoto, Fumie; Katakami, Naoto; et al. (2019). Circulating osteocalcin as a bone-derived hormone is inversely correlated with body fat in patients with type 1 diabetes. PLOS ONE. Dataset. https://plos.figshare.com/articles/dataset/Circulating_osteocalcin_as_a_bone-derived_hormone_is_inversely_correlated_with_body_fat_in_patients_with_type_1_diabetes/8079389/1?file=15057092

The AlOlaiwi2018 diabetes type 2 dataset was collected at the Alwazarat Health Care Center (Riyadh, Saudi Arabia) and is available at figshare: AlOlaiwi, Lina A.; AlHarbi, Turki J.; Tourkmani, Ayla M. (2018). Excel sheet- data analysis dated 25 March 2018 second edition. PLOS ONE. Dataset. https://plos.figshare.com/articles/dataset/Circulating_osteocalcin_as_a_bone_derived_hormone_is_inversely_correlated_with_body_fat_in_patients_with_type_1_diabetes/8079389/1?file=15057092.

Our Python software code is available under the GPL-3.0 license Github and Zenodo:
- https://github.com/gabrielcerono/DiabetesColaboration
- gabrielcerono. (2023). gabrielcerono/DiabetesColaboration: publish (publish). Zenodo. https://doi.org/10.5281/zenodo.8284264

## Supplemental Information

Supplemental information for this article can be found online at http://dx.doi.org/10.7717/peerj-cs.1896#supplemental-information.

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
