# Peer review of "Ensemble machine learning reveals key features for diabetes duration from electronic health records"

_PeerJ Computer Science, doi:10.7717/peerj-cs.1896_

## Round 0.1 · original submission · Major Revisions

Please respond to the reviewers and enrich the manuscript.

·

Basic reporting

The study focused on predicting the past duration of diabetes for patients using machine learning techniques applied to electronic health records. The authors highlight the importance of knowing the duration of diabetes for effective treatment, especially for type 2 diabetes, which commonly occurs in adults. They argue that this information is often missing from patient records, making it challenging for medical professionals to determine the appropriate course of action.
The study utilizes regression analysis and computational intelligence methods on two separate datasets of patients with type 1 and type 2 diabetes. Among the various algorithms employed, Random Forests is found to outperform others in accurately predicting the duration of diabetes for both cohorts. Additionally, the authors employ feature ranking to identify the most relevant factors associated with past diabetes duration, which are found to be age, insulin intake, and body-mass index.
The findings of this study have significant implications for clinical practice. By using the developed tool, medical practitioners can estimate the duration of diabetes even when this information is missing from patient records. This knowledge allows healthcare professionals to make more informed decisions regarding treatment strategies. The highlighted factors—age, insulin intake, and body-mass index—provide valuable insights into the patient's diabetes history, aiding in personalized care and disease management.
Overall, this paper is well-structured and addresses an important issue in diabetes care. The use of machine learning techniques to predict the duration of diabetes based on electronic health records is a promising approach. The identification of age, insulin intake, and body-mass index as key factors further enhances the practical value of the study. However, it should be noted that the abstract lacks specific details on the dataset characteristics, the performance metrics used to evaluate the models, and potential limitations of the study. Including these aspects would provide readers with a more comprehensive understanding of the research conducted.

Experimental design

The methodology not well presented, to improve and strengthen the methodology part, I recommend including a few latest articles related to the topic.
Third major issue is the analysis and results are not enough to be published in such a leading journal. I recommend including a few analyses.
• ML research suffers from many limitations, add a limitation section to addressee various threats to validity of the study should be discussed, e.g., Model evaluation and validation

Validity of the findings

My primary concern about this paper is its suitability to Journal. Although this is an important topic and relevant research, the work seems to fit more into the Computer Science field. One may argue that there is no fine line between Computer Science and Healthcare, and this probably true. However, looking at the details of the paper's background, references, introduction, and conclusion, it is evident to me that the paper serves more health engineering (which is broad, by the way). Indeed, data collection scope is concentrated in healthcare. Therefore, if the authors still have arguments against this concern, I am open to considering their thoughts.
2) To me, an important point in a pure ML paper is its contribution. I see the function of a ML as giving the reader conceptual structuring of the current and previous knowledge on the topic and then giving even more. A good ML study highlights as it’s what knowledge gaps exist, providing also some possible steps for the future research. Further, a good ML study needs to also contribute, not only stating the facts through using already develop model and data set. This contribution can be a model, theory, framework... something that gives the reader a deeper understanding of the phenomenon than what they can get by just reading the lists of facts. Like in any paper, it is important to consider 1) contribution ("what's new?"), 2) impact ("so what?"), 3) logic ("why so?"), and 4) thoroughness ("well done?"). (see e.g. Webster & Watson).
Now, in your paper, you have lots of facts but I'm missing answers to questions of "what's new? so what? and, why so?". I don't think that your contribution is very clear. How much is new in this article compared to the one that addresses already in selected literature papers, would like to see this difference very clearly articulated. It needs to be described more clearly in the paper. I don't see a clear enough additional contribution compared to the published literature.
• I think the discussion of the results about each research objective needs a better summary of the findings. The reader must deal with long discussions about achieved results and several findings from other studies. A more accurate summary (better highlighting findings and lesson learned) could help to better highlight the contribution to the body of knowledge and suggestions that professionals could exploit in their work and researchers in their future research.

Additional comments

Other major issues are:
• It is not clear how you make a data set and from where the data came.
• Even if the goal is interesting as well as the chosen research strategy, I have concerns about designing the research and discussion of results.
• Contributions should be better highlighted.
Minor issues
 The introduction section is too short. Please add problem statement, objective, motivation, and novelty in your study.
• Specifically mentioned the contribution in Introduction.
• English should be improved.
• Summarize literature and mention novelty in the literature review section.
• The methodology does not look like research mythology.
• Support your recommendation with proper literature.
• Mentioned the limitations of the study.

Reviewer 2 ·

Basic reporting

no comment

Experimental design

no comment

Validity of the findings

no comment

Additional comments

1. Line 7 & Line 29: There is a discrepancy between the global number of people with DM mentioned in the Abstract and the information provided on Line 29.

2. Line 70: The statement "In this study, our approach was first to construct a regression model drawing data from 2 different sets of health records" lacks clarity regarding how the author employed a regression model to obtain data.

3. It is recommended to consider merging certain tables, such as Table 2 and Table 3, into a single table or relocating tables and graphs to a supplementary section.

4. Although EHRs typically contain vast data, the working dataset is ~ 400 samples. My guess is the outcome variable is not included in most EHR datasets, however, it would be nice for the authors to clarify this.

5. Regarding the model fitting, I have two questions:
(1) The decision trees applied to the type 2 diabetes data resulted in an R2 of -0.21, indicating a poor fit. Could the author explain this further? Were there any outliers present?
(2) The SMAPE values approaching 50% suggest significant deviations between the true outcomes and the predicted values. It would be helpful if the author could provide a scatter plot with the true outcomes on the x-axis and the predicted values on the y-axis to gain a better understanding of why the predictions were compromised.

---

## Round 0.2 · Major Revisions

Please address the reviewer comments.

·

Basic reporting

The paper titled "Ensemble Machine Learning Reveals Key Features for Diabetes Duration from Electronic Health Records" presents a novel approach to predict diabetes duration by harnessing ensemble machine learning techniques applied to electronic health records (EHRs). Diabetes, a widespread chronic disease, necessitates an understanding of the factors influencing its duration for effective management. This study utilized diverse machine learning models to extract valuable insights from EHR data, uncovering crucial features associated with diabetes duration. The findings have the potential to significantly enhance diabetes management strategies and improve patient outcomes, demonstrating the promise of ensemble machine learning in extracting valuable insights from electronic health records for advancing healthcare research and personalized treatment approaches.

Experimental design

To enhance the experimental design of the study titled "Ensemble Machine Learning Reveals Key Features for Diabetes Duration from Electronic Health Records," several improvements should be considered. These include rigorous data quality control and preprocessing, comprehensive feature selection and engineering, robust data splitting techniques such as time-based splits, expansion of machine learning models with hyperparameter tuning, incorporation of interpretability techniques like SHAP or LIME, external validation across diverse patient populations, strict adherence to ethical and privacy regulations, collaboration with healthcare experts for clinical relevance, and the inclusion of longitudinal analysis to capture the dynamic nature of diabetes duration. By addressing these aspects, the study can provide more robust and clinically actionable insights into diabetes management, benefiting both researchers and healthcare practitioners.

Validity of the findings

To ensure the validity of the findings in the study titled "Ensemble Machine Learning Reveals Key Features for Diabetes Duration from Electronic Health Records," several validation strategies should be considered. These include external validation on independent datasets to assess generalizability, cross-validation techniques to evaluate model stability and generalization, temporal validation to account for the dynamic nature of healthcare data, bootstrap resampling for assessing feature importance consistency, clinical expert review to validate the clinical relevance of key features, interpretability analysis to align findings with medical knowledge, hypothesis testing for statistical significance, robustness testing for modeling variations, and long-term evaluation to confirm the predictions' durability. By implementing these validation approaches, the study can establish the reliability and credibility of its findings, enhancing the utility of its insights for diabetes management and patient care.

Additional comments

Here are some general minor comments and suggestions for improvement:

Clarity and Conciseness: Ensure that the text maintains clarity and conciseness. Some sentences could be further simplified for easier comprehension.

Explicit Details: Include specific details where appropriate. For instance, specify the number of folds in cross-validation and the time period for temporal validation.

Linkage: Make sure there is smooth linkage between sentences and paragraphs to create a coherent flow in the text.

Parallel Structure: Maintain parallel structure in lists or series of suggestions to enhance readability.

Acronyms: When introducing acronyms like SHAP and LIME, consider briefly explaining what they stand for to aid readers who may not be familiar with them.

Real-World Examples: Incorporate real-world examples or case studies if available to illustrate the practical application of the suggested validation strategies.

Emphasize Importance: Emphasize the importance of each validation strategy to underscore why it is crucial for ensuring the validity of the findings.

Consistency in Terminology: Ensure consistent use of terminology throughout the paragraph. For example, if you use "validation" or "assessment," stick with one term for consistency.

Clarity on Pronouns: Ensure that pronouns like "these" and "this" clearly refer to the relevant concepts or strategies.

Concluding Sentence: Consider adding a concluding sentence summarizing the overall significance of these validation strategies for the study.

Grammar and Punctuation: Review the paragraph for any minor grammar or punctuation issues for improved readability.

Remember that these comments are meant to refine and enhance the clarity of the text.

Reviewer 2 ·

Basic reporting

no comment

Experimental design

no comment

Validity of the findings

no comment

Additional comments

1. The prediction of diabetes duration is a captivating subject. Could you provide more insight into how this duration was ascertained from EHR records? From my understanding EHRs are primarily used for reimbursement purposes.

2. The paper suggests that insulin intake is a predictor of diabetes duration. However, if the onset of the disease is unknown, question: why would a patient be administered insulin if they haven't been diagnosed with diabetes? Could you delve deeper into this aspect? Additionally, could you specify the time interval between the first recorded insulin intake and the outcome variables recorded in the EHR?

---

## Round 0.3 · accepted · Accept

I confirm that the authors have addressed all of the reviewers' comments.